# Oral Microbiome of Children Living in an Isolated Area in Myanmar

**DOI:** 10.3390/ijerph17114033

**Published:** 2020-06-05

**Authors:** Yoshiaki Nomura, Ryoko Otsuka, Ryo Hasegawa, Nobuhiro Hanada

**Affiliations:** Department of Translational Research, Tsurumi University School of Dental Medicine, Yokohama 230-8501, Japan; otsuka-ryoko@tsurumi-u.ac.jp (R.O.); hasegawakejp@gmail.com (R.H.); hanada-n@tsurumi-u.ac.jp (N.H.)

**Keywords:** oral microbiome, dental caries, children

## Abstract

Several studies have shown that the oral microbiome is related to systemic health, and a co-relation with several specific diseases has been suggested. The oral microbiome depends on environmental- and community-level factors. In this observational study, the oral microbiomes of children of isolated mountain people were analyzed with respect to the core oral microbiome and etiology of dental caries. We collected samples of supragingival plaque from children (age 9–13) living in the Chin state of Myanmar. After DNA extraction and purification, next-generation sequencing of the V3–V4 hypervariable regions of the 16S rRNA was conducted. From thirteen subjects, 263,458 valid reads and 640 operational taxonomic units were generated at a 97% identity cut-off value. At the phylum level, *Proteobacteria* was the most abundant, followed by *Firmicutes* and *Bacteroides*. Forty-four bacteria were detected in total from all the subjects. For children without dental caries, *Proteobacteria* was abundant. In contrast, in children with dental caries, *Firmicutes* and *Bacteroides* were abundant. The oral microbiome of children living in an isolated area may be affected by environmental- and community-level factors. Additionally, the composition of the oral microbiome may affect the risk of dental caries.

## 1. Introduction

With the advancement of high throughput next-generation sequencing, data on the human oral microbiome has been accumulated. Conventionally, it has been suggested that 700 species of bacteria inhabit the oral cavity [1]. By employing high throughput sequencing, 19,000 phylotypes have been shown to inhabit the oral cavity, including uncultivable bacteria [2]. These bacteria have effects on a person’s health status, especially their oral health. There are some reports that have compared the oral microbiome of subjects with the healthy and diseased oral states: periodontal disease and dental caries [3,4,5]. Additionally, imbalances of oral bacteria have an effect on a person’s systemic health status [6,7,8,9,10,11,12]. There are several ecological niches in the oral cavity, which make the oral microbiome complex [1]. It is well-established that the composition of microbial communities varies in different parts of the oral cavity. The tongue, teeth, mucosa, plate and gingiva have distinctive profiles [13].

In this respect, the concept of a core oral microbiome has been proposed. The core microbiome is longitudinally stable, keeping the human body healthy [14,15]. However, ethnic- or community-level differences in the oral microbial profile have been proposed [16,17].

The prevalence of dental caries has declined across the world. However, it is still prevalent in developing countries. Subjects with rampant caries exist in these countries [1]. Myanmar is a multiracial nation and a developing country with more than 135 ethnic groups. Some of the ethnic groups live in alpine environments in isolation. In these areas, medical and dental supplies and oral care are insufficient. In Myanmar, the prevalence of dental caries is different in urban areas and rural areas. In urban areas, the prevalence of dental caries in children is high: 68.5% in fifth-grade students [18] and 53.2% at the age of 12 [19]. In rural areas, the prevalence of dental caries at the age of 12 has been reported to be 15% [20]. In rural areas, the oral hygiene status is still low; in one study, 61% of children at the age of 12 had never brushed their teeth, and had almost no chance of having sweet snacks or beverages. In rural areas, however, there are subjects with rampant caries [21]. Studies have shown that the overall microbial composition and structure, rather than any particular dominant species, such as *Streptococcus mutans*, can better characterize the cariogenicity of the oral biofilm [22,23,24].

Therefore, information on the oral microbiome of ethnic minorities may be valuable when considering the bacterial etiology of dental caries and ethnic-, environmental- and community-level differences in the oral microbiome. In this study, the oral microbiome of children of mountain people living in the Chin state of Myanmar was analyzed with respect to the core oral microbiome and etiology of dental caries. The aim of this study was to investigate specific oral microbiome profiles and specific species for the risk of dental caries in children living in isolated mountain areas.

## 2. Materials and Methods

### 2.1. Subject

The Government of Myanmar provides mobile and portable dental treatment services in isolated districts. Teams of dental staff from the Naypyidaw National Dental Hospital visit remote places. For children, an oral health checkup and oral hygiene instruction are provided, with local medical staff as translators. Fifty children attended the oral hygiene instruction lecture held at Tonzang on 24 December 2018. 

Children with deciduous teeth were excluded. Thirteen children (9–13 years old) who had more than 15 permanent teeth were included in the analysis. Among them, seven children were caries-free while five children had dental caries. The number of dental caries (D) was D = 2:3, D = 3:1 and D = 4:1, respectively. No subjects had filled teeth or missing teeth due to dental caries.

### 2.2. Oral Examination

An oral examination was carried out at the Tonzang National Hospital. One dentist (R.O.) carried out the oral examination according to the guidelines provided by the World Health Organization. The definition and diagnosis of dental caries were based on the criteria of the World Health Organization [25]. The numbers of decayed teeth, teeth missing due to dental caries and filled teeth were recorded. The sum of decayed, missing and filled teeth was employed as a standardized index (DMF). In this study, no subject had missing or filled teeth, so DMF indicted the number of teeth with untreated dental caries.

### 2.3. Sample Collection

Supragingival plaque samples were collected as previously described [26,27]. Supragingival plaque samples were collected by tooth-brushing for 2 min, followed by immersion of the toothbrush with the attached plaque in sterilized phosphate-buffered saline (PBS). Samples were kept on ice after finishing the examination, and were stored at −20 °C. During transportation, samples were transported in an ice box with a refrigerant.

### 2.4. Microbial DNA Extraction

Dental plaque suspended in PBS was collected by centrifuging the sample at 3000 rpm for 10 min. DNA extraction was performed by the Maxwell 16 LEV Blood DNA Kit (Promega KK, Tokyo, Japan), according to the manufacturer′s instructions. DNA concentrations were measured by Nano Drop ND-2000 (Thermo Fisher Scientific KK, Tokyo, Japan). The degradation of DNA was visually checked by electrophoresis on a 1% agarose gel, and the contamination of RNA was checked using the Qubit dsDNA BR Assay Kit (Thermo Fisher Scientific KK, Tokyo, Japan).

Samples that filled the following criteria were used for further sequence analysis: Concentration >20 ng/μL; volume ≥20 μL; A260/280 ≥ 1.8; and A260/230 > 1.5. In this study, all samples passed these criteria.

### 2.5. Microbial Community Analysis

Extracted DNA was analyzed in the laboratory (Chun Lab, Seoul, Korea). Polymerase chain reaction (PCR) amplification was performed using primers specific to the V3–V4 region next-generation sequencing tags of the 16S rRNA gene in the extracted bacterial DNA. The taxonomic classification of each read was assigned based on a search of the EzBioCloud 16S database [28,29]. This database contains the 16S rRNA genes of strains that have valid published names and representative species-level phylotypes of both cultured and uncultured entries in the GenBank database, with complete hierarchical taxonomic classification from the phylum to the species levels. 

### 2.6. Bioinformatics Analysis

Children were divided into a caries-free group with no dental caries (DMFS = 0) and a caries group (DMFS > 0). The number of 16S rRNA gene copies (absolute abundance) of operational taxonomic units (OTUs) was calculated by multiplying their respective relative abundance by the total number of 16S rRNA gene copies.

For comparison of the two groups, after checking the normality of the values obtained by Kolmogorov–Smirnov tests, Mann Whitney′s U test was applied. To visualize the characteristics of the species in terms of prevalence and abundance, principal component analysis (PCA) was carried out. 

Bioinformatic analysis was performed using the Microbiome package on the Bioconductor of R software (Ver. 3.61).

### 2.7. Ethical Approval

All 50 children who participated in the oral examination were approved for the purpose of this study. Prior to the oral exam, each child or guardian completed an informed consent form. This study was approved by the Ethical Committee of Tsurumi University School of Dental Medicine (Approval Number: 1624).

## 3. Results

From thirteen subjects, 263,458 valid reads were generated. From these valid reads, 640 operational taxonomic units (species) were detected when a 97% identity cut-off value was used.

The alpha diversity indices, Shannon, Simpson, Chao and ACE, were calculated to analyze the diversity and richness of the individual samples. The ACE, Chao1, Jack Knife, Shannon and Simpson were calculated to analyze the diversity and richness of all the samples. When comparing samples of the dental plaque and tongue, the indices of ACE, Chao1, Jack Knife and Shannon were not significantly different (*p* > 0.05), proving that the bacterial diversity and richness were similar in samples collected from the dental plaque and tongue. The mean values of these indices are shown in Appendix A. A rarefaction curve is shown in Appendix A.

Figure 1 shows the relative abundance of the detected bacteria at the phylum level. *Proteobacteria* was the most abundant, followed by *Firmicutes* and *Bacteroides*. Others included *Saccharibacteria* (formerly known as *TM7*), *Spirochetes*, *Synergistetes*, *SR1*, *Peregrinibacteria*, *Tenericutes*, *Chloroflexi* and unclassified organisms in a higher taxonomic rank. 

The heat map constructed from whole reads is shown in Figure 2. By clustering, subjects C5 and H1 were separated; however, seven subjects with dental caries and six subjects with dental caries were separated.

### 3.1. Candidate for the Core Microbiome

From 13 subjects, 44 bacteria were detected. A list of these bacteria is shown in Table 1. These bacteria were candidates for the core microbiome of the human oral cavity; however, some of the pathogenic bacteria were included. 

### 3.2. Difference in the Oral Microbiome between Subjects with or without Dental Caries

Figure 3 shows a comparison of the oral bacterial composition of subjects with or without dental caries. For the subjects without dental caries, *Proteobacteria* was abundant. In contrast, for subjects with dental caries, *Firmicutes* and *Bacteroides* were abundant. The proportions of phyla of each subject that participated in this study are shown in Appendix A. There were four subjects with a proportion of *Proteobacteria* lower than 40% (Sample ID: H6, C2, C3, and C4). For two subjects (Sample ID: H6 and C2), *Bacteroides* was abundant, while for another two subjects (C3 and C4), *Firmicutes* was abundant. These two subjects had dental caries.

The proportions of *Firmicutes* and *Proteobacteria* at the genus level are shown in Appendix A. For *Firmicutes*, organic acid-related bacteria (*Streptococcus* and *Veillonella)* were more prevalent in subjects with dental caries. *Granulicatella*, *Gemella*, *Abiotrophia* and *Bacillus* exhibited a higher prevalence in subjects without dental caries. *Proteobacteria*, *Neisseria*, *Pseudomonas* and *Acinetobactor* were more prevalent in subjects with dental caries, whilst *Aeromonas* was more prevalent in subjects without dental caries.

Table 2 shows the bacteria that displayed statistically significant differences between subjects with or without dental caries. The *Lactobacillus mucosae*, *Neisseria bacilliformis*, *Parascardovia denticolens*, *Prevotella multisaccharivorax*, *Pseudomonas uc* and *Staphylococcus sciuri* groups were not detected in subjects without dental caries. *Veillonella dispar* was not a cariogenic organism. However, it was abundant in the subjects with dental caries. Members of *Veillonella* obtain energy from the utilization of organic acids. Therefore, they may have been isolated from cariogenic circumstances where organic acids were dominant.

The well-known major cariogenic bacteria *Streptococcus mutans* was included; however, *Streptococcus sobrinus* was not included. *Streptococcus sobrinus* was not detected in any of the subjects that participated in this study.

The well-known major cariogenic bacteria *Streptococcus mutans* was more prevalent in subjects with dental caries. However, *Streptococcus sobrinus* was not detected in either group. 

Figure 4 shows the results of the principal component analysis of the significantly different bacteria between subjects with or without dental caries at the species level. *Prevotella multisaccharivorax*, *Neisseria bacilliformis*, *Lactobacillus mucosae* and *Parascardovia denticolens* were located near *Streptococcus mutans*. *Pseudomonas uc* and *Veillonella dispar* were located near each other. For the subjects without dental caries, *Proteobacteria* was abundant. In contrast, for subjects with dental caries, *Firmicutes* and *Bacteroides* were abundant.

## 4. Discussion

In this study, the oral microbiome of 13 children of mountain people living in the Chin state of Myanmar was analyzed, with respect to the core oral microbiome and etiology of dental caries. *Proteobacteria* was the major component of the oral microbiome profile. Several species, including *Streptococcus mutans*, exhibited statistically significant differences in the abundance of dental plaque of children with or without dental caries. 

Several studies have shown that in a healthy oral cavity, 96% of the total oral bacteria can be categorized as *Firmicutes*, *Actinobacteria*, *Proteobacteria*, *Fusobacteria*, *Bacteroides* and *Spirochetes* [15,30,31,32]. The majority of bacteria in a healthy oral cavity is *Firmicutes*, where different species of *Streptococci* are exhibited, with the most abundant being *cocci* [33,34]. The site-specific nature of the oral microbiome has been suggested [35]. In dental plaque, *Firmicutes* and *Actinobacteria* are known to be abundant [2]. In contrast to this, *Proteobacteria* were abundant in this study (Figure 1). *Proteobacteria* is suggested to be abundant in the oral cavity, following *Firmicutes* and *Bacteroidetes* [36]. 

It has been suggested that environmental exposure changes the oral microbiome throughout one’s life [37,38]. Food consumption patterns and contact with exogenous bacteria in food, water, air, other people and domestic animals have been suggested to change the oral microbiome [38,39]. In particular, Western food played a role in the emergence of modern oral pathogens [40]. Oral hygiene habits may play an important role in changing the oral microbiome. The microbial community differs between childhood and adulthood [41,42]. A study which investigated the oral microbiome of infants concluded that the major microbiome consisted of six phyla: *Firmicutes*; *Proteobacteria*; *Actinobacteria*; *Bacteroides*; *Fusobacteria*; and *Spirochaetes* [38]. *Firmicutes* were the most abundant, followed by *Proteobacteria*. 

Another study in Estonia that investigated ninety Estonian schoolchildren (11.3 ± 0.6 years) recorded the presence of *Firmicutes* (39.1%), *Proteobacteria* (26.1%), *Bacteroidetes* (14.7%), *Actinobacteria* (12%) and *Fusobacteria* (6%) [43].

Another study in China investigated 40 young children (age 6–8) with mixed dentition. Seven major phyla (>95% of all sequences) were recorded, including *Firmicutes* (plaque: 27%; saliva: 64%)*, Bacteriodetes* (plaque: 29%; saliva: 13%), *Proteobacteria* (plaque: 18%; saliva: 11%), *Fusobacteria* (plaque: 19%; saliva: 3%), *Actinobacteria* (plaque: 3%; saliva: 8%), *Saccharibacteria* (TM7) (plaque: 1.5%; saliva: 0.5%) and *Spirochaetes* (plaque: 0.59%; saliva: 0.065%)*. Firmicutes*, *Bacteroides* and *Fusobacteria* occupied more than 70% of the oral microbiome of the subjects with mixed dentition. Additionally, *Proteobacteria* had a prevalence of less than 20% [44]. Our results showed that *Proteobacteria* was the most abundant bacteria in our sample set, due to ethnic- or isolated community-level differences investigated in previous reports [45,46]. 

The proportion of *Firmicutes* was higher in children with dental caries (Figure 3). *Streptococcus mutans* and lactobacilli belonging to the phylum *Firmicutes* were major cariogenic bacteria [47]. 

According to the results of the principal component analysis, the species *Lactobacillus mucosae, Neisseria bacilliformis*, *Parascardovia denticolens* and *Prevotella multisaccharivorax* were located near *S. mutans* (Figure 4). Furthermore, these four species were detected in subjects with dental caries and not detected in subjects without dental caries (Table 2). Therefore, there is a possibility that they have a nature similar to the cariogenic bacteria *S. mutans.*

The existence of restorations affects oral microbiome profiles. It is well-known that resin monomers have a biological effect on human immune system cells [48], and may disturb the oral microbiome profile. In this study, however, no children had restated teeth or orthodontic appliances in their oral cavity. Pit and fissure sealant is an effective tool for the prevention of dental caries [49]. Dental materials employed for the prevention of dental caries also disturb oral microbiome profiles. Materials used for pit and fissure sealant contain resin monomer and fluoride, and fluoride affects streptococci [50]. In the mountainous area of this study, many children do not use toothpaste, and there are no fluoride mouth rinse prevention programs. Additionally, no children had pit and fissure sealant. In this study, dental materials may have little or no effect on the microbiome profile.

Diet plays an important role in dental caries. *S. mutans* produces lactic acid under anaerobic conditions. In addition, other acidic metabolites, such as formate and acetate ethanol, are produced [51]. Under this environment, acidophilic bacteria such as lactobacilli favor growth in the oral cavity. Acid production by an excess uptake of carbohydrates results in dental caries [52]. The carbohydrate-rich modern diet may act as an environmental factor, causing changes that increase the cariogenicity of the oral microbiome. In this study, the oral microbiome of the children with dental caries showed a similar pattern to the microbiome of adults in other studies [37]. 

The limitations of this study were the small sample size, and the fact that only dental plaque samples were analyzed. Moreover, we only investigated one area of Myanmar, where more than 150 ethnic groups live. As the daily food intake affects the microbiome, nutrients of local food should be analyzed and compared with the oral microbiome. Defining the core microbiome requires further studies to investigate a wider ethnic group and larger sample size.

## 5. Conclusions

In the oral microbiome of the children living in the isolated area of this study, the most abundant phylum was *Proteobacteria.* The oral microbiome may be affected by environmental- and community-level factors. The oral microbiomes were different between children with or without dental caries. The composition of the oral microbiome may thus affect the risk of dental caries.

## Figures and Tables

**Figure 1 ijerph-17-04033-f001:**
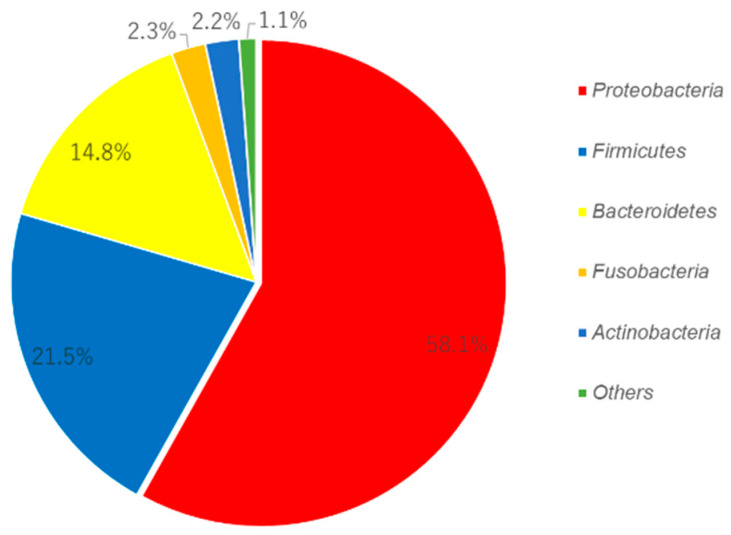
Abundance of the oral microbiome of the 13 subjects at the phylum level.

**Figure 2 ijerph-17-04033-f002:**
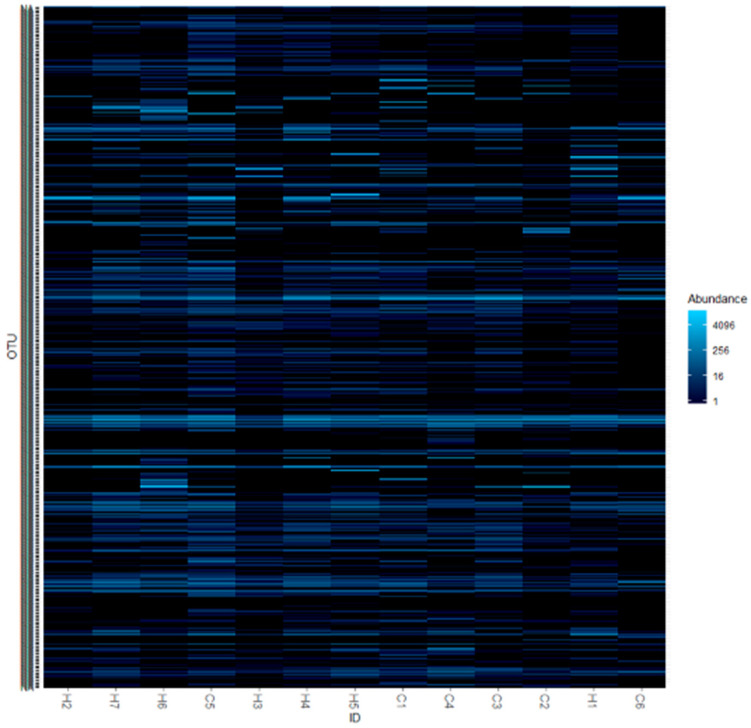
Heat map constructed from whole reads from 13 subjects.

**Figure 3 ijerph-17-04033-f003:**
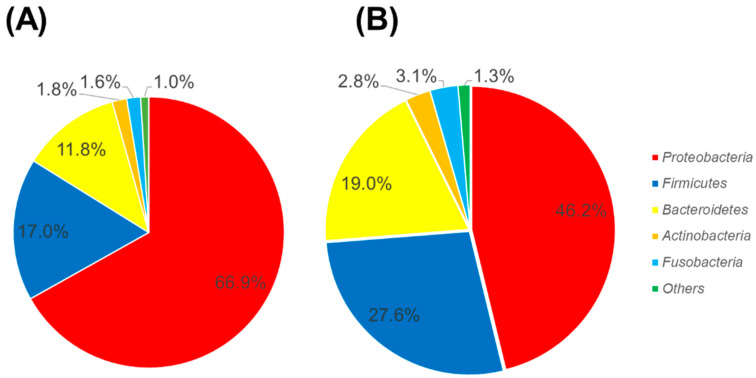
Comparison of the oral bacterial composition of subjects with or without dental caries. (**A**) Subjects without dental caries, and (**B**) subjects with dental caries.

**Figure 4 ijerph-17-04033-f004:**
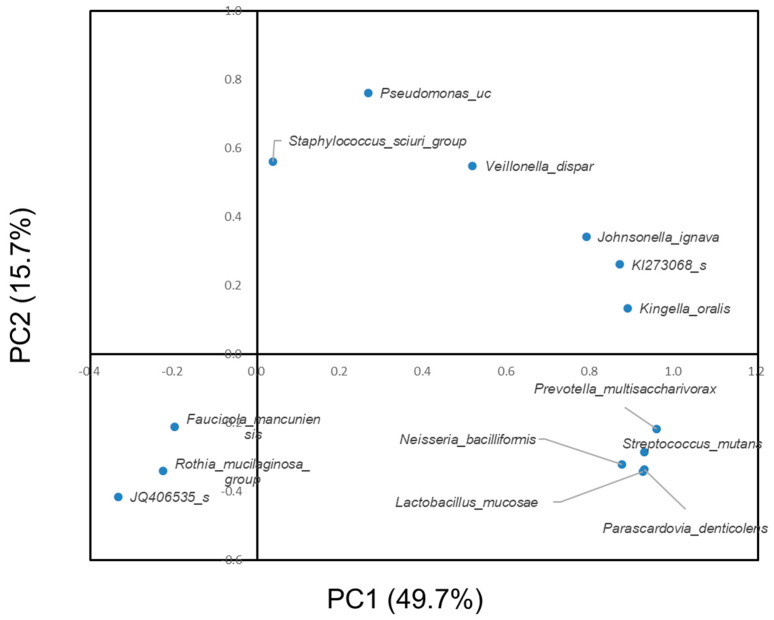
Principal component analysis of the significantly different bacteria between subjects with or without dental caries at the species level.

**Table 1 ijerph-17-04033-t001:** A list of the bacteria found in samples from all thirteen subjects.

Taxon Name	Abundance (%)
*Veillonella parvula* group	5.22% (0.48–21.92%)
*Neisseria sicca* group	4.72% (0.09–31.89%)
*Streptococcus pneumoniae* group	4.40% (0.37–8.80%)
*Haemophilus parainfluenzae* group	3.60% (0.21–8.34%)
*Lautropia mirabilis*	2.99% (0.27–9.79%)
*Streptococcus sanguinis* group	1.89% (0.18–4.00%)
*Veillonella dispar*	1.78% (0.03–7.47%)
*Streptococcus parasanguinis* group	1.24% (0.15–2.89%)
*Granulicatella adiacens* group	0.99% (0.05–3.57%)
*Aggregatibacter aphrophilus*	0.92% (0.02–5.49%)
*Fusobacterium nucleatum* group	0.74% (0.07–1.55%)
*Veillonella rogosae*	0.72% (0.02–3.17%)
*Porphyromonas pasteri*	0.72% (0.01–2.35%)
*Streptococcus peroris* group	0.61% (0.03–1.30%)
*Gemella morbillorum*	0.57% (0.02–2.05%)
*Leptotrichia buccalis* group	0.54% (0.02–1.97%)
*Aggregatibacter segnis*	0.54% (0.01–1.63%)
*Capnocytophaga granulosa*	0.47% (0.04–1.49%)
*Streptococcus gordonii* group	0.46% (0.04–2.15%)
*Prevotella loescheii*	0.46% (0.03–2.12%)
*Abiotrophia defectiva*	0.41% (0.01–1.40%)
*Capnocytophaga sputigena*	0.39% (0.02–1.27%)
*KI259256_s*	0.36% (0.02–1.90%)
*Streptococcus_uc*	0.36% (0.01–0.90%)
*Streptococcus sinensis* group	0.34% (0.03–1.07%)
*ADCM_s*	0.31% (0.02–0.54%)
*JQ463704_s*	0.28% (0.02–1.14%)
*Gemella haemolysans* group	0.25% (0.02–0.90%)
*Veillonella_uc*	0.24% (0.01–0.90%)
*CP017038_s*	0.22% (0.03–0.53%)
*Campylobacter gracilis*	0.21% (0.02–0.65%)
*Cardiobacterium hominis*	0.19% (0.02–0.40%)
JF239777_s	0.18% (0–0.98%)
Unclassified in a higher taxonomic rank	0.16% (0.03–0.45%)
*Streptococcus anginosus* group	0.16% (0.02–0.65%)
*Prevotella oris*	0.12% (0.01–0.50%)
*Actinomyces odontolyticus*	0.11% (0.01–0.47%)
*Corynebacterium matruchotii*	0.09% (0.01–0.51%)
*Dialister invisus*	0.09% (0.01–0.27%)
*Actinomyces oris*	0.08% (<0.01–0.29%)
*Granulicatella elegans*	0.07% (<0.01–0.23%)
*Campylobacter concisus* group	0.07% (0.02–0.16%)
*Actinomyces naeslundii*	0.06% (<0.01–0.21%)
*Prevotella maculosa*	0.06% (<0.01–0.13%)

**Table 2 ijerph-17-04033-t002:** The significantly different bacteria between subjects with or without dental caries at the species level.

	Dental Caries				*p*-Value
	−		+		
OTUs	Mean ± SD	Median (25th–75th%)	Mean ± SD	Median (25th–75th%)	
*Faucicola mancuniensis*	0.4136 ± 1.0826	0.0043 (0–0.0181)	-		0.036
*Johnsonella ignava*	0.0019 ± 0.0051	0 (0–0)	0.0082 ± 0.00686	0.0029 (0.0029–0.0156)	0.030
*JQ406535 s*	0.0321 ± 0.0435	0.0136 (0–0.048)	-		0.015
*KI273068 s*			0.0043 ± 0.00539	0.0019 (0–0.0104)	0.042
*Kingella oralis*	0.0037 ± 0.0063	0 (0–0.0045)	0.0360 ± 0.0276	0.0319 (0.0095–0.0627)	0.006
*Lactobacillus mucosae*	-		0.0563 ± 0.12536	0.0049 (0–0.0900)	0.042
*Neisseria bacilliformis*	-		0.0083 ± 0.01179	0.0036 (0–0.0168)	0.042
*Parascardovia denticolens*	-		0.0217 ± 0.04818	0.0025 (0–0.0340)	0.042
*Prevotella multisaccharivorax*	-		0.0535 ± 0.09038	0.0208 (0–0.0927)	0.015
*Pseudomonas uc*	-		0.4038 ± 0.87311	0.0255 (0–0.6889)	0.015
*Rothia mucilaginosa* group	0.0469 ± 0.0660	0.0175 (0.0086–0.0815)	0.0038 ± 0.00585	0 (0–0.011)	0.030
*Staphylococcus sciuri* group	-		0.0039 ± 0.00459	0.0025 (0–0.0084)	0.042
*Streptococcus mutans*	0.0062 ± 0.0050	0.0086 (0–0.0099)	0.0960 ± 0.1302	0.0670 (0.0093–0.1455)	0.031
*Veillonella dispar*	0.4027 ± 0.6048	0.1981 (0.0481–0.5183)	3.3924 ± 2.6177	2.9546 (0.9421–5.8599)	0.007

- No dental caries; + With dental caries.

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
