# Peer review of "Oral Microbiome of Children Living in an Isolated Area in Myanmar"

_ijerph, 2020, doi:10.3390/ijerph17114033_

Round 1

Reviewer 1 Report

It was extremely difficult to follow the arguments as the language was very difficult to follow. I think this needs to be improved in the first instance before the manuscript can be reasonably reviewed and evaluated on its scientific merits. 

Author Response

Response to Reviewers

Reviewer 1

Thank you for the valuable comments to improve our manuscript (manuscript ID: ijerph-809302).

It was extremely difficult to follow the arguments as the language was very difficult to follow. I think this needs to be improved in the first instance before the manuscript can be reasonably reviewed and evaluated on its scientific merits.

<Response>

We used English editing service provided by MDPI. We hope the manuscript improved.

Reviewer 2 Report

The authors have gone to an isolated area in Myanmar and sampled the oral microbiome of 13 children and contrasted the 16S rRNA gene sequences from a group that did not have caries with one that did. They also examined the sequences to define a core oral microbiome. Sampling isolated groups is important to help our understanding of how the oral microbiome changes between groups of people. However, the manuscript needs some serious rewriting and the authors need to consider finding a native English speaker/writer to assist them with their narrative. Readers will have difficulty understanding the way this is written.

Other points.

Abstract, Line 14 (and other places in the manuscript, e.g., Line 78). The authors state that they did pyrosequencing to get the 16S rRNA gene sequences. Pyrosequencing was one of the first methods devised for next generation sequencing and has a specific mechanism. However, the company that was supporting it (Roche) discontinued this support in 2013. So, the authors need to make sure that 'pyrosequencing' was actually used. Most investigations use Illumina sequencing to get their data and the fact that these authors used primers for the V3-V4 region of the 16S rRNA gene to get sequences leads to the suspicion that Illumina sequencing was actually used here and the authors are confusing the 'pyrosequencing' with 'next generation sequencing'. This is important to aany reader of this manuscript.

Introduction, Line 30 (and in the reference list), The authors cite [6,7] in the text, but in the reference list, these are both exactly the same. Are they really different or should there only be one citation?

Materials and Methods, Lines 84 and 85, The term DMFS is used for the first time in the manuscript. For readers who don't know what this is, the authors need to spell this out completely the first time it is used. 

More information is needed on statistical analyses (e.g. PCA).

Results, Lines 125-131, All the species names should be written in italics.

Discussion. The authors should consider presenting their major findings first before indicating what other studies have shown. They should also consider whether the small number of individuals (13) provides some limitations to how their research can be viewed in a larger context. They also make minimal use of their own figures and tables in the Discussion. If they are writing about their own data, they need to let the reader know about which figure or table the data comes from.

Conclusions. The authors need to list their specific conclusions supported from their data.

Author Response

Response to Reviewers

Reviewer 2

Thank you for the valuable comments to improve our manuscript (manuscript ID: ijerph-809302). We replied your valuable comments point by point. Changes were high lightened yellow in revised manuscript.

  • However, the manuscript needs some serious rewriting and the authors need to consider finding a native English speaker/writer to assist them with their narrative. Readers will have difficulty understanding the way this is written.

<Response>

We used English editing service provided by MDPI. We hope the manuscript improved.

  • Abstract, Line 14 (and other places in the manuscript, e.g., Line 78). The authors state that they did pyrosequencing to get the 16S rRNA gene sequences. Pyrosequencing was one of the first methods devised for next generation sequencing and has a specific mechanism. However, the company that was supporting it (Roche) discontinued this support in 2013. So, the authors need to make sure that 'pyrosequencing' was actually used. Most investigations use Illumina sequencing to get their data and the fact that these authors used primers for the V3-V4 region of the 16S rRNA gene to get sequences leads to the suspicion that Illumina sequencing was actually used here and the authors are confusing the 'pyrosequencing' with 'next generation sequencing'. This is important to any reader of this manuscript.

<Response>

We found three pyrosequencing in Abstract, Method and Conclusion. These words were replaced to pyrosequencing.

Others were title of papers in Reference.

  • Introduction, Line 30 (and in the reference list), The authors cite [6,7] in the text, but in the reference list, these are both exactly the same. Are they really different or should there only be one citation?

<Response>

Thank you for pointing out our mistake. Reference 7 was removed.

  • Materials and Methods, Lines 84 and 85, The term DMFS is used for the first time in the manuscript. For readers who don't know what this is, the authors need to spell this out completely the first time it is used.

<Response>

Following paragraph was inserted in the “2.2 oral examination”

<Inserted paragraph>

The numbers of decayed teeth, teeth missing due to dental caries, and filled teeth were recorded. The sum of the decayed, missing, and filled teeth was employed as a standardized index (DMF). In this study, no subject had missing or filled teeth, so DMF indicted the number of teeth with untreated dental caries.

  • More information is needed on statistical analyses (e.g. PCA).

<Response>

Following paragraph was inserted in 2.6 Bioinformatics analysis.

<Inserted paragraph>

For comparison of the two groups, after checking the normality of the values obtained by Kolmogorov–Smirnov tests, Mann Whitney’s U test was applied. To visualize the characteristics of the species in terms of prevalence and abundance, principal component analysis (PCA) was carried out.

  • Results, Lines 125-131, All the species names should be written in italics.

<Response>

Corrected. Names were written Italic.

  • The authors should consider presenting their major findings first before indicating what other studies have shown.

<Response>

Findings of this study was inserted in the first paragraph in Discussion.

Following paragraph was inserted in the discussion.

<Inserted paragraph>

Proteobacteria was the major component of the oral microbiome profile. Several species, including Streptococcus mutans, exhibited statistically significant differences in the abundance of dental plaque of children with or without dental caries.

  • They should also consider whether the small number of individuals (13) provides some limitations to how their research can be viewed in a larger context. They also make minimal use of their own figures and tables in the Discussion.

<Response>

Following paragraph was inserted in Discussion as limitation of this study.

<Inserted paragraph>

The limitations of this study were the small sample size and the fact that only dental plaque samples were analyzed. Moreover, we only investigated one area of Myanmar, whilst there are more than 150 ethnic groups living in the country. As the daily food intake affects the microbiome, nutrients of local food should be analyzed and compared with the oral microbiome.

  • If they are writing about their own data, they need to let the reader know about which figure or table the data comes from.

<Response>

 “Table” and “Fig” were inserted as index.

  • The authors need to list their specific conclusions supported from their data.

<Response>

Following sentence was inserted in conclusion.

<Inserted sentence>

In the oral microbiome of the children living in the isolated area of this study, the most abundant phylum was Proteobacteria.

Reviewer 3 Report

Thank you for submitting your study. 

Major concerns with the study include:

  1. Study population - N = 13 is a very low number. Why weren't more participants included? What is the statistical value of a study population of 13.
  2. What is the efficiency of collecting the microbiota on the toothbrushes? How was the sample collection method validated?
  3. Were the molecular methodologies validated and correlated with conventional microbiology?

Author Response

Response to Reviewers

Reviewer 3

Thank you for the valuable comments to improve our manuscript (manuscript ID: ijerph-809302). We replied your valuable comments point by point. Changes were high lightened yellow in revised
manuscript.

Study population - N = 13 is a very low number. Why weren't more participants included? What is the statistical value of a study population of 13.

<Response>

Various age of children attended oral examinations. We thought that analysis by children with only deciduous teeth and only permanent tooth made the results confusing. In this study, children with deciduous teeth was excluded.

For the sample size calculation, statistical significance is obtained by large sample size, there is a risk of type one error. Statistical significance obtained by small sample size, there is little risk f type one error. There exist clear difference n the population.

Following sentence was inserted in 2.1 Subjects in Method.
<Inserted paragraph>
Children with deciduous teeth were excluded. Thirteen children (9-13 years old) who had more than 15 permanent teeth were included in the analysis. Among them, seven children were caries-free and five children had dental caries. The number of dental caries (D) was D = 2:3, D = 3:1, and D = 4:1, respectively. No subjects had filled teeth or missing teeth due to dental caries.

What is the efficiency of collecting the microbiota on the toothbrushes? How was the sample collection method validated?

<Response>

We first begin this series of study by chimpanzee [1-6]. Sampling from chimpanzee is very limited. Under general anesthesia, we collected plaque samples by brushing tooth surface. To compare the data, plaque samples have been used. Then, the concern about representativeness of plaque sample was
raised. Study to compare the presence of specific bacteria in plaque, saliva and periodontal pocket was carried out [7]. The results indicated almost common results were obtained by that plaque and saliva. Sensitivity of samples from periodontal pocket was low. We did not compare the sample of tongue or
fornix. Because, under the clinical practice of dental treatment, saliva, plaque or gingival cervical fluid have been used for the risk assessment of dental caries or periodontal disease. Therefore, under the limited condition including this study, only plaque sample are used. Further study is necessary to compare the data from tongue or fornix. It is the limitation of this study. Therefore, following sentence was included as limitation of this study.

<Inserted paragraph>
The limitations of this study were the small sample size and the fact that only dental plaque samples were analyzed. Moreover, we only investigated one area of Myanmar, whilst there are more than 150 ethnic groups living in the country. As the daily food intake affects the microbiome, nutrients of local food should
be analyzed and compared with the oral microbiome.

1. Okamoto M, Imai S, Miyanohara M, Saito W, Momoi Y, Nomura Y, Ikawa T, Ogawa T, Miyabe-Nishiwaki T, Kaneko A, Watanabe A, Watanabe S, Hayashi M, Tomonaga M, Hanada N.Streptococcus panodentis sp. nov. from the oral cavities of chimpanzees. Microbiol Immunol. 2015 Sep;59(9):526-32. doi: 10.1111/1348-0421.12290
2. Okamoto M, Naito M, Miyanohara M, Imai S, Nomura Y, Saito W, Momoi Y, Takada K, Miyabe-Nishiwaki T, Tomonaga M, Hanada N. Complete genome sequence of Streptococcus troglodytae TKU31 isolated from the oral cavity of a chimpanzee (Pantroglodytes). Microbiol Immunol. 2016 Dec;60(12):811-816. doi: 10.1111/1348-0421.12453.
3. A Novel Mutans Streptococci isolated from Chimpanzee Oral Cavity. IADR Division: Pan European Region Meeting, Meeting: 2012 Pan European Region Meeting (Helisinki, Finland)
4. Biofilm formation and demineralization by mutans streptococci from human and chimpanzee. IADR Division: Japanese Division Meeting Meeting: 2015 Japanese Division Meeting (Fukuoka, Japan)
5. Distribution of Streptococci in Oral Flora of Chimpanzee. IADR Division: Pan European Region Meeting Meeting: 2012 Pan European Region Meeting (Helisinki, Finland).
6. Pyrosequencing Analysis of Oral Flora isolated from Chimpanzees. IADR Division: Continental European and Scandinavian Divisions Meeting Meeting: 2011 Continental European and Scandinavian Divisions Meeting (Budapest, Hungary)
7. Okada A, Sogabe K, Takeuchi H, Okamoto M, Nomura Y, Hanada N.Characterization of specimens obtained by different sampling methods for evaluation of periodontal bacteria. J Oral Sci. 2017 Dec 27;59(4):491-498. doi: 10.2334/josnusd.16-0573. Epub 2017 Nov 17.
Were the molecular methodologies validated and correlated with conventional microbiology?
<Response>
The molecular methodologies sued in this study was outsourcing for Chun Lab in Korea. They construct EzBioCloud Database for taxonomic assignment [1]. The methodology depended on their techniques.
1. Introducing EzBioCloud: a taxonomically united database of 16S rRNA gene sequences and whole-genome assemblies.
Int J Syst Evol Microbiol. 2017 May; 67(5): 1613–1617. PMID: 28005526 PMCID: PMC5563544

Reviewer 4 Report

This is an interesting work on the definition of the composition parameters of the oral microbiome in children of the Myanmar region.

However, some criticisms are present:

-Abstract: an initial sentence on the role of the oral microbiome in relation to systemic diseases should be inserted

-Abstract: Line 11 insert the nature of the study (observational? ...)

-Introduction: some general considerations on the specificity of the oral microbiome must be absolutely inserted. In particular, emphasis should be placed on the presence of at least three ecological niches unlike other body districts

-Line 30 The part of the introduction that recalls the possible relationships between oral microbiome and systemic pathologies is absolutely insufficient. Some recent studies relate it to reflux esophagitis, autism, gastrointestinal problems; report these aspects recalling them from the scientific literature.

-Line 34: the authors should provide more accurate epidemiological data on the development of carious disease in the study region. In particular, I expect a distinction between age groups, education level, etc.

-Line 35 Insert bibliographic reference to support the sentence

-Line 45 At the end of the introduction section, the null hypotheses must be inserted.

-Line 45: better specify the objectives of the study

-Line 54: the inclusion and exclusion criteria of the study were not specified

-Line 54: report the possible approval of the Ethics Committee for the execution of the study

-Line 57: insert in detail, adding a specific table, the phases of the oral examination with particular reference to the methods of analysis

-Line 64 Remove the point after examination

- Line 60 how did you deal with sampling, with reference to the intake of food or drinks before detection?

-Line 65 Why was only the supragingival plaque sample selected and not also the tongue or fornix sample? Insert bibliographic references to support the choice

-Line 83: the choice of caries free or carious groups must be reported at the beginning of the materials and methods section. What were the study group selection criteria? What minimum number of carious lesions was chosen?

-Line 144: why hasn't Alpha diversity been performed to study the results achieved? It is absolutely necessary to insert it

-Line 165: the authors speak of studies but indicate only the reference number 36

-In the discussion section another aspect is not considered, namely the possible relationship between oral dysbiosis and the presence of dental restorations. In this regard, I recommend inserting the following scientific work in the reference section which could be helpful in understanding the text:

1: Pagano S, Rabbit M, Valenti C, Negri P, Lombardo G, Costanzi E, Cianetti S,Montaseri A, Marinucci L. Biological effects of resin monomers on oral cell populations: descriptive analysis of literature. Eur J Paediatr Dent. 2019

Sep; 20 (3): 224-232

- Another important aspect to consider concerns the potential clinical effects of defining a microbiome that predisposes to dental caries.

- In particular, the development of prevention protocols in more predisposed subjects, for example with oral sealants, would be very important, especially in regions such as that considered by authors to be difficult to access to dental structures. In this regard, I recommend inserting the following scientific work in support of the discussion in the reference section:

- Bromo, F., Guida, A., Santoro, G., Peciarolo, MR, Eramo, S. (2011) Pit and fissure sealants: review of literature and application technique. (Review) Minerva stomatologica 60 (10): 529 -541

Author Response

Response to Reviewers

Reviewer 4

Thank you for the valuable comments to improve our manuscript (manuscript ID: ijerph-809302). We replied your valuable comments point by point. Changes were high lightened yellow in revised manuscript.

  • -Abstract: an initial sentence on the role of the oral microbiome in relation to systemic diseases should be inserted

<Response>

Following sentences were inserted.

<Inserted sentence>

Several studies have shown that the oral microbiome is related to systemic health, and a co-relation with several specific diseases has been suggested.

  • -Abstract: Line 11 insert the nature of the study (observational? ...)

<Response>

Cross sectional or observational is appropriate. “Observational” was inserted.

<Previous manuscript>

In this study, oral microbiome of the children of isolated...

<Revised manuscript>

In this observational study, the oral microbiomes of children of isolated mountain people were analyzed with respect to the core oral microbiome and etiology of dental caries.

  • -Introduction: some general considerations on the specificity of the oral microbiome must be absolutely inserted. In particular, emphasis should be placed on the presence of at least three ecological niches unlike other body districts

<Response>

Following sentence was inserted.

<Inserted sentence>

There are several ecological niches in the oral cavity, which make the oral microbiome complex [1]. It is well-established that the composition of microbial communities varies in different parts of the oral cavity. The tongue, teeth, mucosa, plate, and gingiva have distinctive profiles [13].

  • -Line 30 The part of the introduction that recalls the possible relationships between oral microbiome and systemic pathologies is absolutely insufficient. Some recent studies relate it to reflux esophagitis, autism, gastrointestinal problems; report these aspects recalling them from the scientific literature.

<Response>

References were added Ref: 7-12

  • -Line 34: the authors should provide more accurate epidemiological data on the development of carious disease in the study region. In particular, I expect a distinction between age groups, education level, etc.

<Response>

Following paragraph was inserted in Introduction.

<Inserted paragraph>

In Myanmar, the prevalence of dental caries is different in urban areas and rural areas. In urban areas, the prevalence of dental caries in children is high: 68.5% in fifth-grade students [18] and 53.2% at the age of 12 [19]. In rural areas, the prevalence of dental caries at the age of 12 has been reported to be 15% [20]. In rural areas, the oral hygiene status is still low; in one study, 61% of children at the age of 12 had never brushed their teeth. They had almost no chance of having sweet snacks or beverages.

  • -Line 35 Insert bibliographic reference to support the sentence

<Response>

Reference was inserted. Ref 1.

  • -Line 45 At the end of the introduction section, the null hypotheses must be inserted.
  • -Line 45: better specify the objectives of the study

<Response>

Following sentence was inserted in the last of Introduction.

<Inserted sentence>

The aim of this study was to investigate specific oral microbiome profiles and specific species for the risk of dental caries in children living in isolated mountain areas.

  • -Line 54: report the possible approval of the Ethics Committee for the execution of the study

<Response>

Following section was inserted.

2.8.Ethical Approval

Informed consent was obtained from each child before the oral examination. This study was approved by the Ethical Committee of Tsurumi University School of Dental Medicine (Approval Number: 1624).

  • -Line 57: insert in detail, adding a specific table, the phases of the oral examination with particular reference to the methods of analysis

<Response>

We used the attached standardized assessment form. The analysis is only sum up decayed, filled and missing teeth. Actually there was no filled teeth and missing teeth.

https://www.who.int/oral_health/publications/9789241548649/en/

  • -Line 64 Remove the point after examination

<Response>

Thank you for pointing out our mistake. Grammar error was revised.

  • Line 60 how did you deal with sampling, with reference to the intake of food or drinks before detection?

<Response>

This oral health promotion event was announced for the resident by leaflet and poster. The announcement contained oral health checkup and oral health instruction for children. For the oral health checkups was held at 10 AM. We advised to finish breakfast until 8.AM. We checked it by interview. After that oral health instruction was carried out.

  • -Line 65 Why was only the supragingival plaque sample selected and not also the tongue or fornix sample? Insert bibliographic references to support the choice

<Response>

We first begin this series of study by chimpanzee[1-6]. Sampling from chimpanzee is very limited. Under general anesthesia, we collected plaque samples by brushing tooth surface. To compare the data, plaque samples have been used. Then, the concern about representativeness of plaque sample was raised. Study to compare the presence of specific bacteria in plaque, saliva and periodontal pocket was carried out [7]. The results indicated almost common results were obtained by that plaque and saliva. Sensitivity of samples from periodontal pocket was low. We did not compare the sample of tongue or fornix. Because, under the clinical practice of dental treatment, saliva, plaque or gingival cervical fluid have been used for the risk assessment of dental caries or periodontal disease. Therefore, under the limited condition including this study, only plaque sample are used. Further study is necessary to compare the data from tongue or fornix. It is the limitation of this study. Therefore, following sentence was included as limitation of this study.

<Inserted paragraph>

The limitations of this study were the small sample size and the fact that only dental plaque samples were analyzed. Moreover, we only investigated one area of Myanmar, whilst there are more than 150 ethnic groups living in the country. As the daily food intake affects the microbiome, nutrients of local food should be analyzed and compared with the oral microbiome.

  1. Okamoto M, Imai S, Miyanohara M, Saito W, Momoi Y, Nomura Y, Ikawa T, Ogawa T, Miyabe-Nishiwaki T, Kaneko A, Watanabe A, Watanabe S, Hayashi M, Tomonaga M, Hanada N.Streptococcus panodentis sp. nov. from the oral cavities of chimpanzees. Microbiol Immunol. 2015 Sep;59(9):526-32. doi: 10.1111/1348-0421.12290
  2. Okamoto M, Naito M, Miyanohara M, Imai S, Nomura Y, Saito W, Momoi Y, Takada K, Miyabe-Nishiwaki T, Tomonaga M, Hanada N. Complete genome sequence of Streptococcus troglodytae TKU31 isolated from the oral cavity of a chimpanzee (Pan troglodytes). Microbiol Immunol. 2016 Dec;60(12):811-816. doi: 10.1111/1348-0421.12453.
  3. A Novel Mutans Streptococci isolated from Chimpanzee Oral Cavity. IADR Division: Pan European Region Meeting, Meeting: 2012 Pan European Region Meeting (Helisinki, Finland)
  4. Biofilm formation and demineralization by mutans streptococci from human and chimpanzee. IADR Division: Japanese Division Meeting Meeting: 2015 Japanese Division Meeting (Fukuoka, Japan)
  5. Distribution of Streptococci in Oral Flora of Chimpanzee. IADR Division: Pan European Region Meeting Meeting: 2012 Pan European Region Meeting (Helisinki, Finland).
  6. Pyrosequencing Analysis of Oral Flora isolated from Chimpanzees. IADR Division: Continental European and Scandinavian Divisions Meeting Meeting: 2011 Continental European and Scandinavian Divisions Meeting (Budapest, Hungary)
  7. Okada A, Sogabe K, Takeuchi H, Okamoto M, Nomura Y, Hanada N.Characterization of specimens obtained by different sampling methods for evaluation of periodontal bacteria. J Oral Sci. 2017 Dec 27;59(4):491-498. doi: 10.2334/josnusd.16-0573. Epub 2017 Nov 17.

  • -Line 54: the inclusion and exclusion criteria of the study were not specified
  • -Line 83: the choice of caries free or carious groups must be reported at the beginning of the materials and methods section. What were the study group selection criteria? What minimum number of carious lesions was chosen?

<Response>

Various age of children attended oral examinations. We thought that analysis by children with only deciduous teeth and only permanent tooth made the results confusing. In this study, children with deciduous teeth was excluded.

Following sentence was inserted in 2.1 Subjects in Method.

<Inserted paragraph>

Children with deciduous teeth were excluded. Thirteen children (9-13 years old) who had more than 15 permanent teeth were included in the analysis. Among them, seven children were caries-free and five children had dental caries. The number of dental caries (D) was D = 2:3, D = 3:1, and D = 4:1, respectively. No subjects had filled teeth or missing teeth due to dental caries.

  • -Line 144: why hasn't Alpha diversity been performed to study the results achieved? It is absolutely necessary to insert it

<Response>

 The results of Alpha diversity was included.

The alpha diversity indices: Shannon, Simpson, Chao, and ACE were calculated to analyze the diversity and richness of the individual samples. The ACE, Chao1, Jack Knife, Shannon, and Simpson were calculated to analyze the diversity and richness of all the samples. When comparing samples of the dental plaque and tongue, the indices of ACE, Chao1, Jack Knife, and Shannon were not significantly different (P > 0.05), proving that the bacterial diversity and richness were similar in samples collected from the dental plaque and tongue. The mean values of these indices are shown in Table S1. A rarefaction curve is shown in Figure S2.

  • -Line 165: the authors speak of studies but indicate only the reference number 36

<Response>

Error was revised.

<Previous manuscript>

Studies investigated the oral microbiome……….

<Revised manuscript>

A study investigated the oral microbiome……….

  • -In the discussion section another aspect is not considered, namely the possible relationship between oral dysbiosis and the presence of dental restorations. In this regard, I recommend inserting the following scientific work in the reference section which could be helpful in understanding the text:

1: Pagano S, Rabbit M, Valenti C, Negri P, Lombardo G, Costanzi E, Cianetti S,Montaseri A, Marinucci L. Biological effects of resin monomers on oral cell populations: descriptive analysis of literature. Eur J Paediatr Dent. 2019 Sep; 20 (3): 224-232

  • Another important aspect to consider concerns the potential clinical effects of defining a microbiome that predisposes to dental caries. In particular, the development of prevention protocols in more predisposed subjects, for example with oral sealants, would be very important, especially in regions such as that considered by authors to be difficult to access to dental structures. In this regard, I recommend inserting the following scientific work in support of the discussion in the reference section:

Bromo, F., Guida, A., Santoro, G., Peciarolo, MR, Eramo, S. (2011) Pit and fissure sealants: review of literature and application technique. (Review) Minerva stomatologica 60 (10): 529 -541

<Response>

In this study, no children had filled teeth. No children had no restorations. No children experienced orthodontic treatment.

Following paragraph was inserted in Discussion.

<Inserted paragraph>

The existence of restorations affects oral microbiome profiles. It is well-known that resin monomers have biological effects on human immune system cells [50]. They may disturb the oral microbiome profile. However, in this study, no children had restated teeth and no children had orthodontic appliances in their oral cavity. Pit and fissure sealant is an effective tool for the prevention of dental caries [51]. Dental materials employed for the prevention of dental caries also disturb oral microbiome profiles. Materials used for pit and fissure sealant contain resin monomer and fluoride, and fluoride affects streptococci [52]. In the mountainous area of this study, many children do not use tooth paste and there are no fluoride mouth rinse prevention programs. Additionally, no children had pit and fissure sealant.

  1. Pagano, S.; Rabbit, M.; Valenti, C.; Negri, P.; Lombardo, G.; Costanzi, E.; Cianetti, S.;Montaseri A.; Marinucci, L. Biological effects of resin monomers on oral cell populations: descriptive analysis of literature. Eur J Paediatr Dent. 2019 20, 224-232. PMID: 31489823 DOI: 10.23804/ejpd.2019.20.03.11
  2. Bromo F.; Guida A.; Santoro G.; Peciarolo M R.; Eramo S. Pit and Fissure Sealants: Review of Literature and Application Technique. Minerva Stomatol. 2011, 60, 529-541. PMID: 22082857
  3. Bunick, F.J.; Kashket, S. Enolases from fluoride-sensitive and fluoride-resistant streptococci. Infect Immun. 1981 Dec;34(3):856-63. PMID: 7333671 PMCID: PMC350948

Round 2

Reviewer 1 Report

The  manuscript has been improved considerably.

On the ethical aspects. The authors indicate that informed consent was obtained from the children who are minors.  

Were the parents and or guardians approached for written  consent.

Did the children all assent? 

Author Response

  • On the ethical aspects. The authors indicate that informed consent was obtained from the children who are minors.
  • Were the parents and or guardians approached for written consent.
  • Did the children all assent?

Response

The manuscript was revived.  Line 117-118

<Previous manuscript>

Informed consent was obtained from each child before the oral examination.

<Revised manuscript>

All 50 children who participated in the oral examination were approved for the purpose of this study. Prior to the oral exam, each child or guardian completed an informed consent form.

Reviewer 2 Report

The authors have satisfactorily made the changes/corrections to previous comments. A sentence in the Discussion from line 238 to 239 appears to be unfinished. This should be corrected.

Author Response

A sentence in the Discussion from line 238 to 239 appears to be unfinished. This should be corrected.

 Response

Following sentence was inserted.

Line 238-239

In this study, dental materials may have little or no effect on the microbiome profile.

Reviewer 3 Report

  • Study should be considered  a "pilot study" given the low number of participants.
    •  Please describe follow on plans for expanding the study.
  •  Not clear how children can sign an informed consent. Please explain.

Author Response

Study should be considered a "pilot study" given the low number of participants.

Please describe follow on plans for expanding the study.

Response

Following sentence was inserted in Discussion.

Line: 250-251

Defining the core microbiome requires further study to investigate a wider ethnic group and larger sample size.

Not clear how children can sign an informed consent. Please explain.

Response

The manuscript was revised. Line 117-118

<Previous manuscript>

Informed consent was obtained from each child before the oral examination.

<Revised manuscript>

All 50 children who participated in the oral examination were approved for the purpose of this study. Prior to the oral exam, each child or guardian completed an informed consent form.

Reviewer 4 Report

all questions were modified

i reccomend work acceptation after english grammar revision

Author Response

I recommend work acceptation after English grammar revision.

Response

This manuscript has almost no change after MDPI English editing.